# Three-Dimensional Analysis of Isolated Orbital Floor Fractures Pre- and Post-Reconstruction with Standard Titanium Meshes and “Hybrid” Patient-Specific Implants

**DOI:** 10.3390/jcm9051579

**Published:** 2020-05-22

**Authors:** Guido R. Sigron, Nathalie Rüedi, Frédérique Chammartin, Simon Meyer, Bilal Msallem, Christoph Kunz, Florian M. Thieringer

**Affiliations:** 1Department of Oral and Cranio-Maxillofacial Surgery, University Hospital Basel, CH-4031 Basel, Switzerland; guido.sigron@usb.ch (G.R.S.); nathalie.rueedi@unibas.ch (N.R.); simon.meyer@usb.ch (S.M.); bilal.msallem@usb.ch (B.M.); christoph.kunz@usb.ch (C.K.); 2Medical Additive Manufacturing Research Group (Swiss MAM), Department of Biomedical Engineering, University of Basel, CH-4123 Allschwil, Switzerland; 3Basel Institute for Clinical Epidemiology and Biostatistics, Department of Clinical Research, University Hospital Basel, University of Basel, CH-4031 Basel, Switzerland; frederiquesophie.chammartin-basnet@usb.ch

**Keywords:** orbital fracture, blow-out fracture, orbital volume, hybrid patient-specific, orbital implant, 3D-printing

## Abstract

The aim of this study was to compare the efficacy of the intraoperative bending of titanium mesh with the efficacy of pre-contoured “hybrid” patient-specific titanium mesh for the surgical repair of isolated orbital floor fractures. In-house 3D-printed anatomical models were used as bending guides. The main outcome measures were preoperative and postoperative orbital volume and surgery time. We performed a retrospective cohort study including 22 patients who had undergone surgery between May 2016 and November 2018. The first twelve patients underwent conventional reconstruction with intraoperative free-hand bending of an orbital floor mesh plate. The subsequent ten patients received pre-contoured plates based on 3D-printed orbital models that were produced by mirroring the non-fractured orbit of the patient using a medical imaging software. We compared the preoperative and postoperative absolute volume difference (unfractured orbit, fractured orbit), the fracture area, the fracture collapse, and the effective surgery time between the two groups. In comparison to the intraoperative bending of titanium mesh, the application of preformed plates based on a 3D-printed orbital model resulted in a non-significant absolute volume difference in the intervention group (*p* = 0.276) and statistically significant volume difference in the conventional group (*p* = 0.002). Further, there was a significant reduction of the surgery time (57.3 ± 23.4 min versus 99.8 ± 28.9 min, *p* = 0.001). The results of this study suggest that the use of 3D-printed orbital models leads to a more accurate reconstruction and a time reduction during surgery.

## 1. Introduction

Orbital fractures are among the most frequent fractures of the midface and may cause severe functional impairment [1]. The repair of orbital floor fractures is challenging because of the complexity of the anatomical region involved and the limited intraoperative view [2]. Fractures involving the orbital walls can result in a change of the orbital volume and alter the position of the eye. The goal of orbital reconstruction is the accurate repair of the premorbid orbital form and function [3,4]. Moreover, the reconstruction should be performed as gently as possible in order not to additionally damage soft tissue. Due to the extremely challenging architecture, inaccurate surgical techniques may lead to clinical failures such as diplopia, enophthalmos, and even vision loss [5].

Advances in 3D-printing technologies enable accurate preoperative planning of orbital reconstructions. Modern, increasingly cost-effective technologies are more widely available for clinical use and enable the creation of patient-specific implants [6,7]. The combination of 3D imaging and individually adjusted so-called “hybrid patient-specific” titanium meshes have recently been introduced for effective reconstruction of the orbit. Pre-bent implants with a 3D-printed prototype of an orbital model have been found to be more accurate than conventional, freehand bent plates for the reconstruction of orbital fractures [8,9,10,11,12]. Compared to directly printed patient-specific titanium implants, which are usually manufactured externally and are still time- and cost-intensive, the so-called hybrid implants represent an interesting customized alternative.

The aim of this retrospective cohort study was to compare the efficacy of the intraoperative bending of titanium mesh with a hybrid patient-specific titanium mesh, preformed based on an in-house 3D-printed anatomical model, for the surgical repair of isolated orbital floor fractures, with preoperative and postoperative orbital volume and surgery time as main outcome measures.

## 2. Experimental Section

Between May 2016 and November 2018, a total of 30 patients with unilateral isolated orbital floor fractures underwent a surgical reconstruction with orbital floor mesh at the department for oral and maxillofacial surgery of the University Hospital Basel in Switzerland. The study protocol was approved by the Ethics Committee of Northwest and Central Switzerland (EKNZ; Project-ID 2019-00260). Only patients with both complete clinical records, preoperative and postoperative computed tomography (CT), or cone beam computed tomography (CBCT) scans, and ophthalmic examinations were included in this study. Twenty-two patients (10 women and 12 men) met the inclusion criteria. The patients were divided into two treatment groups (Figure 1).

The first 12 patients underwent conventional reconstruction with intraoperative freehand bending of an orbital floor mesh plate (MatrixMIDFACE, DePuy Synthes, Solothurn, Switzerland). The ten subsequent patients received pre-bent plates (MatrixMIDFACE, DePuy Synthes, Solothurn, Switzerland or MODUS Midface OPS 1.5, Medartis, Basel, Switzerland) that were precontoured with equipment provided by the department based on a 3D-printed orbital model by mirroring the non-fractured orbit of the patient using the CE-certified medical software Mimics Innovation Suite v. 20–21 (Materialise, Leuven, Belgium), and two different desktop 3D-printer, a MakerBot Replicator+ (MakerBot Industries, Brooklyn, NY, USA) and an Objet30 Prime (Stratasys, Ltd., Eden Prairie, MN, USA). The orbital floor mesh plate was trimmed, if needed, and hand-molded by a resident specializing in maxillofacial surgery to fit the size of the defect according to the 3D printed orbital model (Figure 2). For intraoperative use, the pre-bent plate was sterilized in an autoclave with a standardized and certified sterilization procedure. During the entire precontouring process, the printed 3D model never had contact with the patient. In addition, we used commercially available and medically approved orbital meshes, not directly 3D-printed patient-specific implants (PSI).

The clinical records were retrospectively screened for sex, age, mechanism of injury, and latency time between trauma and surgery (Table 1). All fractures were classified using a semiquantitative method described by Kunz et al. and the Arbeitsgemeinschaft für Osteosynthesefragen—Craniomaxillofacial Surgery (AO CMF) craniomaxillofacial classification system, supported by the AO comprehensive injury automatic classifier software (AO COIAC) software [13,14]. We compared the preoperative and postoperative orbital volume, the fracture area, and the maximum fracture collapse between the two groups (Table 2). Furthermore, to assess the efficacy of orbital reconstruction with pre-bent plates, the surgical time was also compared between the two groups.

### 2.1. Surgical Procedure

Under general anesthesia, a senior maxillofacial surgeon and a resident performed mid-eyelid, transconjunctival, and/or transcaruncular approaches to expose the orbital wall defects. After the reduction of herniated orbital tissue, the freehand bent (group 1) or pre-bent (group 2) orbital floor mesh plates were placed and fixated with titanium screws to the inferior orbital rim. Before direct wound closure with cutaneous sutures, a forced duction test was performed to confirm the unrestricted passive movements of the eye.

### 2.2. Semi-Automatic Orbital Fracture Analysis

The preoperative and postoperative orbital volume, the fracture area, and the maximum fracture collapse were calculated semi-automatically with the Disior Bonelogic CMF Orbital software v. 0.4.1r (Disior Ltd., Helsinki, Finland). For this, preoperative and postoperative CT (Siemens Somatom, Erlangen, Germany) or CBCT (Carestream CS 9300, Rochester, NY, USA) images were imported in digital imaging and communications in medicine (DICOM) format into the software. After import, the software automatically converts the image information into a voxel-map and creates a 3D rendering of craniofacial bone structure with a specified Hounsfield unit (HU) value of 300. After rendering, one observer defined a point (one-click method) approximating the posterior closure of orbital apex (Figure 3).

This point represents the anterior edge of the optic nerve foramen opening, at the level of its greatest anterior–posterior length. Then, the software numerically analyses fracture and orbital dimensions by an iterative expansion and deformation of a ball-shaped sphere and different algorithms.

### 2.3. Statistical Analysis

All data analyses were performed using the R program v. 3.6.1 (Foundation for Statistical Computing, Vienna, Austria). The results were presented as mean, standard deviation, or range, if not indicated otherwise, and the significance of differences in means was examined using Student’s two-sample t-test and Fisher’s exact test. Statistical significance was determined at *p* < 0.05.

## 3. Results

The 22 included patients (10 women, 12 men) had a mean age of 49.8 years (range 20–83 years). The mean time from trauma to surgical repair in the conventional group was 4.1 ± 3.1 days and in the intervention group was 2.8 ± 2.5 days. The descriptive statistics of the patients are shown in Table 1.

The mean (SD) orbital volume of the preoperative non-fractured orbits in the conventional group was 31.6 (4.2) mL and 26.1 (2.2) mL in the intervention group (*p* = 0.001). The mean (SD) volume of the preoperative fractured orbits was 33.1 (4.7) mL in the conventional group and 28.4 (4.0) mL in the intervention group (*p* = 0.020). The mean (SD) volume of the reconstructed orbits was 30.1 (4.2) mL in the conventional group and 25.7 (3.0) mL in the intervention group (*p* = 0.010).

The mean (SD) absolute volume difference (volume of preoperative non-fractured orbit minus volume of reconstructed orbit) in the conventional group was 1.6 (1.2) mL, whereas in the intervention group it was 1.0 (0.7) mL. This results in a statistically significant difference in the conventional group (*p* = 0.002); in contrast, in the intervention group there was no significant difference noted between the mean absolute volume difference of the non-fractured orbits and the reconstructed orbits (Figure 4).

The mean (SD) value of the preoperative fractured area was 408.5 (137.5) mm^2^ in the conventional group and 389.4 (135.1) mm^2^ in the intervention group; this difference was not statistically significant. The mean (SD) maximum fracture collapse was 6.9 (2.3) mm in the conventional group and 8.6 (5.4) mm in the interventional group; this difference was also not statistically significant. All these measurements are presented in Table 2.

Mean (SD) surgery time (from the point of skin incision to the end of skin closure) was statistically significantly (*p* = 0.001) shorter in the intervention group (57.3 (23.4) min) than in the conventional group (99.8 (28.9) min). There was no significant difference regarding the mean (SD) post-operative length of stay in hospital between the conventional group (3.8 (3.0) days) and the intervention group (4.6 (3.9) days) (Figure 5).

## 4. Discussion

Orbital floor fractures are associated with potential complications like diplopia, enophthalmos, and infraorbital and optical nerve injuries due to nearby sensitive structures such as the eyeball, optic nerve, and extraocular muscles [15]. Innovative technologies like computer-assisted approaches have taken a key role during the past few years, allowing more precise and safer surgery [16]. Recently, for unilateral orbital defects, the mirroring of the unaffected orbit was proven to be a legitimate reference for orbital reconstruction [17,18]. Three-dimensional-printing provides an accurate anatomical representation for this purpose and allows to preoperatively adapt a patient-specific plate for orbital reconstruction [6,19,20]. The total time required for segmenting DICOM data and preparing the data for 3D-printing usually takes less than 1 h, including post-processing (the removal of support material).

It was shown that precise volumetric reconstruction could be achieved using an individually manufactured rapid prototype skull model and a pre-bent synthetic scaffold [21] (Figure 6).

Our method of measuring orbital volume has proven to be effective and accurate (Figure 7). Thus, in this study, the mean preoperative and reconstructed orbital volumes were similar to those found in earlier studies [17,22]. Furthermore, we were able to show that the application of pre-bent titanium meshes leads to a more accurate orbital volume reconstruction than freehand bent titanium meshes.

Earlier studies of orbital reconstructions using preformed titanium meshes based on 3D-printed orbital models have also postulated that patient-specific implants provide precise results and are therefore superior to manually bent titanium mesh [23]. In addition, customized plates have been shown to reduce the amount of manipulation of tissue [24].

One of the main advantages of pre-bent implants based on 3D-printed orbital models is the significant reduction of surgery time. Our results regarding this are in line with the results of previous studies that could show that individual planning decreases the operation time [24,25]. In the study of Zieliński et al., a shorter time of operation means lower intraoperative blood loss and expected shorter hospitalization time [26]. In contrast, our study showed no difference between length of stay between the two groups.

In our department using pre-bent implants can considerably reduce operation time by 42.5 min. The estimated costs for the surgical theatre add up to $47.50–$103 per minute in Switzerland [27,28]. Therefore, the money saved resulting from the multiplication of time gain by cost per minute in our study was up to $4,377.50. Of course, costs vary greatly depending on staffing levels, hospital infrastructure costs, and, ultimately, the healthcare system of the different countries.

The material cost of one 3D-printed orbital model in our study is estimated at $2 for PLA (MakerBot Replicator+) and $25 for Med610 (Objet30 Prime). The aforementioned virtual planning of a 3D orbital model took approximately 20 min and was usually performed by a surgical resident. The image segmentation work and fabrication of the 3D model was appreciated by the residents as preparation time for the surgical procedure.

This study has certain limitations. First, the sample size as small; but this had no impact on the comparison of the two methods. Second, the study was based on a retrospective design with a possible selection bias. The risk of a selection bias as minimal, as we have included all consecutive patients who were treated for isolated orbital floor fracture in our department.

## 5. Conclusions

The results of this study indicate that the preoperative fabrication of 3D-printed anatomical orbital models to pre-contour hybrid patient-specific implants can be a valuable and cost-effective tool for the reconstruction of orbital wall fractures. The use of pre-bent, patient-specific hybrid titanium meshes significantly reduces surgery time and results in at least equally accurate orbital volume reconstruction compared to a freehand bent titanium mesh.

We believe that semi-automated image segmentation and registration for volume analysis and surface visualization is a milestone for the automated manufacturing of patient-specific implants. As soon as technology, medico-legal regulations and approval for use of these models allow, in a few years, the immediate printing of implants at the point-of-care will play an essential role in the treatment of facial skull fractures. These methods should be considered for further studies.

## Figures and Tables

**Figure 1 jcm-09-01579-f001:**
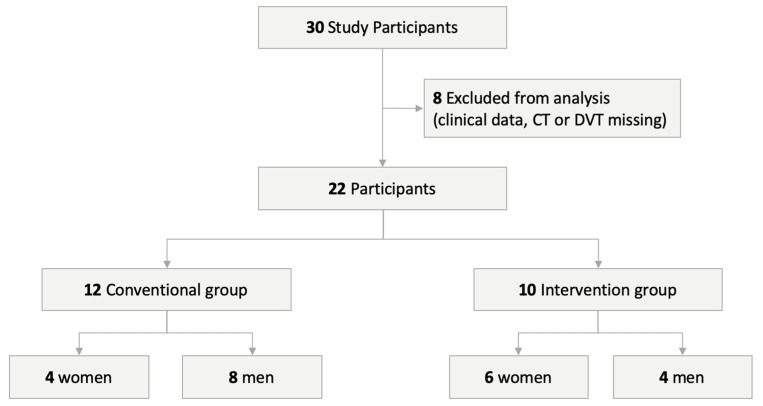
Visualization of the study design.

**Figure 2 jcm-09-01579-f002:**
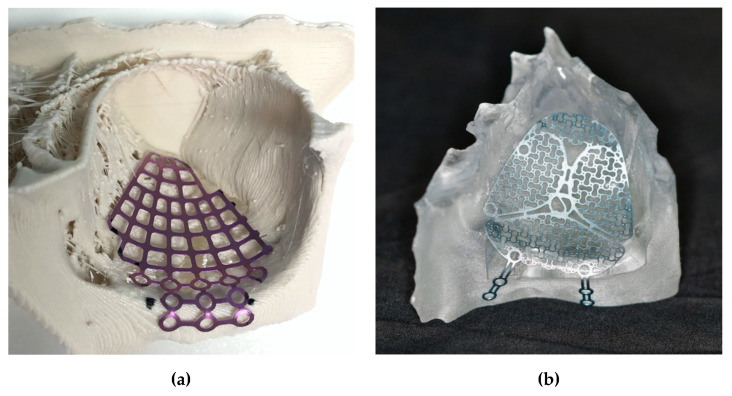
Pre-bent plates based on a 3D-printed orbital model: (**a**) Orbital floor mesh plate MatrixMIDFACE on a polylactic acid (PLA) model printed with a MakerBot Replicator+; (**b**) Medartis Modus Midface OPS 1.5 plate on a biocompatible clear MED610 model printed with a Stratasys Objet30 Prime.

**Figure 3 jcm-09-01579-f003:**
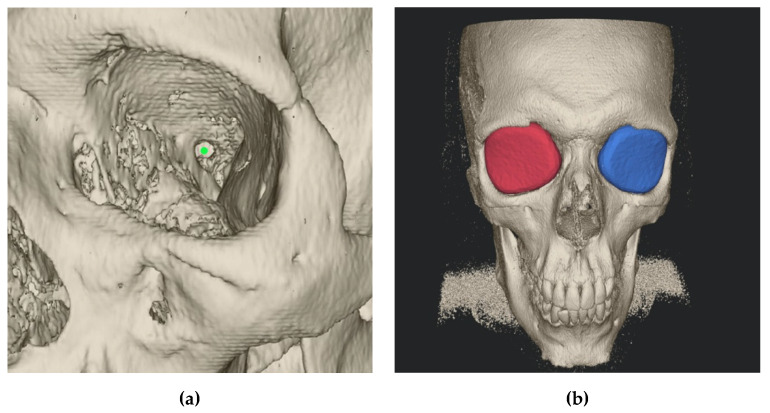
Disior Bonelogic CMF Orbital software (Craniomaxillofacial surgery) software: (**a**) Rendered craniofacial bone with defined point (green bullet) at the anterior edge of the optic nerve foramen opening; (**b**) Orbital volume analyses by an iteratively expansion and deformation of a ball-shaped sphere.

**Figure 4 jcm-09-01579-f004:**
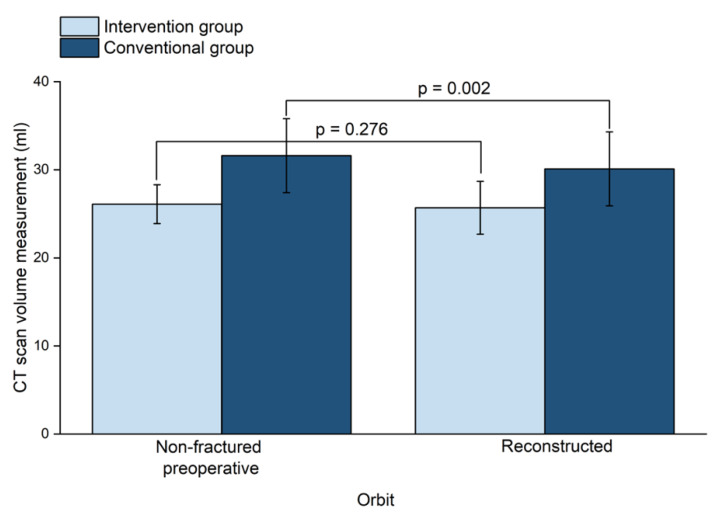
The volume of non-fractured orbits and reconstructed orbits between the two groups. CT: Computed tomography.

**Figure 5 jcm-09-01579-f005:**
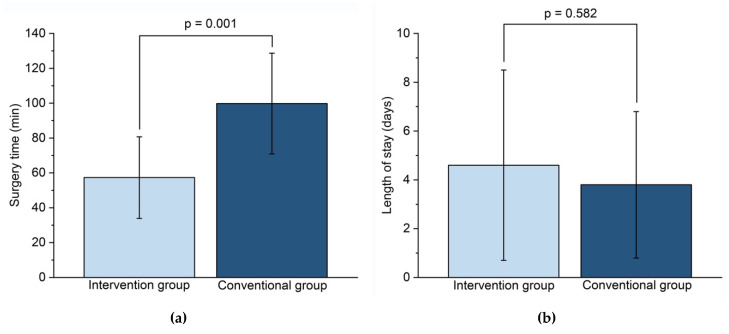
Comparison between intervention and conventional group: (**a**) Surgery time (min); (**b**) Length of stay (days).

**Figure 6 jcm-09-01579-f006:**
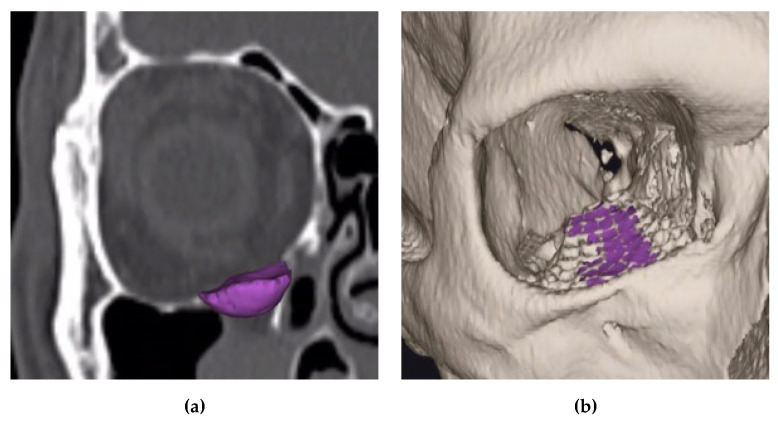
Disior Bonelogic CMF Orbital software: (**a**) Three-dimensional reconstruction of the orbital floor defect in the area of the right orbit (purple); (**b**) Former defect area (purple) after reconstruction in projection on the titanium mesh.

**Figure 7 jcm-09-01579-f007:**
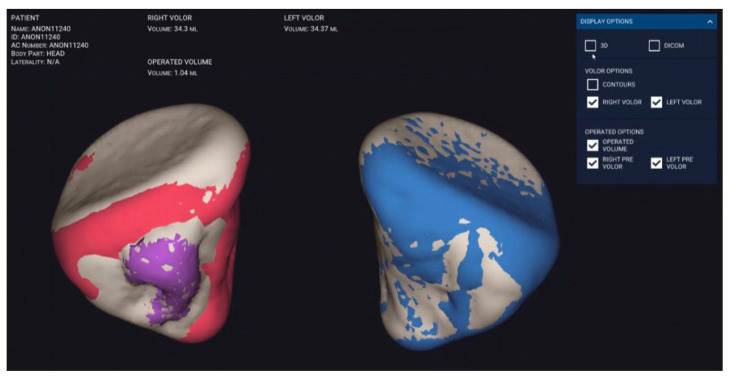
Disior Bonelogic CMF Orbital software: Automatically segmented orbit (right, in red with defect region in purple; and left, intact in blue).

**Table 1 jcm-09-01579-t001:** The descriptive statistics of the study sample.

Variable	Intervention Group	Conventional Group
**Number of Patients (%)**	10 (45.45%)	12 (54.55%)
**Age in Years, Mean (Range)**	47.5 (20–83)	51.8 (21–79)
Sex		
Male	4	8
Female	6	4
Cause of Injury		
Fall	5	7
Assaults	2	4
Sports Injuries	2	1
Work-Related Injuries	1	0
Days Between Trauma and Surgery	2.8 ± 2.5	4.1 ± 3.1
Surgery Time (Min)	57.3 ± 23.4	99.8 ± 28.9
Length of Stay (Days)	4.6 ± 3.9	3.8 ± 3.0
Fracture Classification		
AO CMF		
92 m.OiW2(i)	5	10
92 Oi.mW2(i)	5	2
Kunz et al.		
A1	3	7
A2	6	5
A3	1	0

AO CMF: Arbeitsgemeinschaft für Osteosynthesefragen—Craniomaxillofacial Surgery, A1: Isolated defect of the orbital floor, or the medial wall, 1–2 cm^2^, A2: Defect of the orbital floor in the anterior two-third, or the medial wall, or both, >2 cm^2^, bony ledge preserved at medial margin of the infraorbital fissure, A3: Defect of the orbital floor in the anterior two-third, or the medial wall, or both, >2 cm^2^, missing bony ledge medial to the infraorbital fissure.

**Table 2 jcm-09-01579-t002:** Preoperative and postoperative measurements.

	Non-Fractured Orbit (mL)	Fractured orbit (mL)	
**Conventional Group**			
Preoperative	31.6 ± 4.2	33.1 ± 4.7	
Postoperative	31.4 ± 4.3	30.1 ± 4.2	
Absolut Volume Difference (mL)			1.6 ± 1.2 *
Fracture Area (mm^2^)			408.5 ± 137.5
Fracture Max. Collapse (mm)			6.9 ± 2.3
**Intervention Group**			
Preoperative	26.1 ± 2.2	28.4 ± 4.0	
Postoperative	26.1 ± 2.2	25.7 ± 3.0	
Absolut Volume Difference (mL)			1.0 ± 0.7
Fracture Area (mm^2^)			389.4 ± 135.1
Fracture Max. Collapse (mm)			8.6 ± 5.4

* Statistically significant *p* = 0.002.

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
