# Peer review of "Three-Dimensional Analysis of Isolated Orbital Floor Fractures Pre- and Post-Reconstruction with Standard Titanium Meshes and “Hybrid” Patient-Specific Implants"

_jcm, 2020, doi:10.3390/jcm9051579_

Round 1

Reviewer 1 Report

The article is of interest to ENT community and should be published with some minor corrections.

There is  some objections about cost effectiveness of the procedure described by the authors using 3D pre-bent implants and the money saved by the technology.

This can be true only in the country like Switzerland where the cost of OR is one of the highest in the world. The cost of OR per minute showed in the paper are tremendous…. . In majority of other countries when the cost of medical treatment is singificantly lower , the use of this technology will be much more expensive that traditional,  due to considerable cost of equipment for 3D printing. So the overall cost of the described hybrid procedure will be much higher that conventional treatment with a routine - mannually bent titanium mesh .

Thus the sentence about cost effectivness and money saved should be changed and should take under consideration my comments. The cost effectiveness depends mostly on where it is performed.

It does not decrease the value of the paper of course.

There is also some inconsequence between the abstact and the discussion.

In abstract the authors state that the use of both techniques reconstructing the orbit resulted in accurate restoration with statistically non-significant absolute volume differences but reduction of surgery time, and in the discussion they state, that they were able to show that the application of pre-bent titanium meshes leads to a more accurate orbital volume reconstruction than freehand bent titanium meshes???? The authors are obliged to be consequent and reffer to statisticall results in their statements.. As I understand the main benefit was the time reduction during surgery and of course it is a very important achievement…. .

Author Response

Manuscript ID: jcm-787345

“Three-dimensional analysis of isolated orbital floor fractures pre- and post-reconstruction with standard titanium meshes and “hybrid” patient-specific implants”

At the very beginning the authors would like to express their sincere appreciation to the editor and the reviewer for their valuable contributions to the manuscript. The authors have taken all the input into account and have amended the manuscript. Below you will find the detailed answers to all comments and suggestions of the reviewers. The authors would be pleased if the changes made in the revised manuscript were considered satisfactory for publication.

Reviewer 1

The article is of interest to ENT community and should be published with some minor corrections.

  • Comment 1:

There is  some objections about cost effectiveness of the procedure described by the authors using 3D pre-bent implants and the money saved by the technology.

This can be true only in the country like Switzerland where the cost of OR is one of the highest in the world. The cost of OR per minute showed in the paper are tremendous…. . In majority of other countries when the cost of medical treatment is singificantly lower , the use of this technology will be much more expensive that traditional,  due to considerable cost of equipment for 3D printing. So the overall cost of the described hybrid procedure will be much higher that conventional treatment with a routine - mannually bent titanium mesh .

Thus the sentence about cost effectivness and money saved should be changed and should take under consideration my comments. The cost effectiveness depends mostly on where it is performed.

It does not decrease the value of the paper of course.

Author Response:

Thank you very much for your valuable comments. The authors agree that it is necessary to change the sentence about cost effectiveness. The cost per minute actually depends mainly on the hospital set up and country where the surgery is performed. Furthermore, we agree and value the comment that in the majority of other countries, when the cost of medical treatment is significantly lower, the use of this technology can be significantly more expensive than the traditional method. On the other hand, 3D printers and consumables no longer cost much money. This fact has also led to the adoption of low-cost 3D printing systems for anatomical model production in virtually every healthcare system around the world.

Correction in the revised manuscript:

Line 199-202: The estimated costs for the surgical theatre add up to $47.50 - $103 per minute in Switzerland [27,28]. Therefore, the money saved resulting from the multiplication of time gain by cost per minute in our study was up to $4,377.50. Of course, costs vary greatly depending on staffing levels, hospital infrastructure costs and ultimately the health care system of the different countries.

  • Comment 2:

There is also some inconsequence between the abstact and the discussion. 

In abstract the authors state that the use of both techniques reconstructing the orbit resulted in accurate restoration with statistically non-significant absolute volume differences but reduction of surgery time, and in the discussion they state, that they were able to show that the application of pre-bent titanium meshes leads to a more accurate orbital volume reconstruction than freehand bent titanium meshes???? The authors are obliged to be consequent and reffer to statisticall results in their statements.. As I understand the main benefit was the time reduction during surgery and of course it is a very important achievement…. .

Author Response:

The authors acknowledge the comment and agreed on this inconsequence between the abstract and the discussion. Therefore the manuscript has been corrected to rectify the error that was made in the abstract section.

Correction in the revised manuscript:

Line 31-37: In comparison to the intraoperative bending of titanium mesh, the application of preformed plates based on a 3D-printed orbital model resulted in a non-significant absolute volume difference in the intervention group (p = 0.276) and statistically significant volume difference in the conventional group (p = 0.002). Further, there is a significant reduction of the surgery time (57.3 ± 23.4 min versus 99.8 ± 28.9 min, p = 0.001). The results of this study suggest that the use of 3D-printed orbital models leads to a more accurate reconstructions and a time reduction during surgery.

Line 144-149: The mean (SD) absolute volume difference (volume of preoperative non-fractured orbit minus volume of reconstructed orbit) in the conventional group was 1.6 (1.2) ml, whereas in the intervention group 1.0 (0.7) ml. This results in a statistically significant difference in the conventional group (p = 0.002), in contrast in the intervention group there was no significant difference noted between the mean absolute volume difference of the non-fractured orbits and the reconstructed orbits (Fig. 4).

Reviewer 2 Report

This is a small, retrospective study that the authors use to build off of their prior studies.  This study suggests that results are at least comparable to free hand mesh bending.

Who performed the surgeries and who bent the plates? trainees? faculty? one faculty vs multiple faculty?

Do you have federal/institutional approval for use of in-house printed models for patient use (i.e. bending of plates) aside from ethics committee approval? Authors do mention briefly in the conclusions medico-legal implications for in-house custom implant printing but do not mention approval for use of these models.

Authors discuss cost for saved operative time - I would like to see cost estimates for the printing, materials, and time of the person printing (is this 1 hour of physician time?) and compare the differences.

This study discussed trying to improve cost-effectiveness of care by using a hybrid pre-bent implant. However, post-operative CT scans or cone beam CT scans were obtained in patients in the study.  This is not standard in many institutions and increases cost of care and radiation exposure. 

Authors also mention potential for decreased blood loss and shorter hospitalization time (line 185-6).  However, repair of isolated orbital floor fractures are typically outpatient procedures with minimal blood loss.

Although small study , retrospective, with multiple limitations, does discuss a "hot topic" for using 3-D printing and custom implants in CMF trauma.

Author Response

Manuscript ID: jcm-787345

“Three-dimensional analysis of isolated orbital floor fractures pre- and post-reconstruction with standard titanium meshes and “hybrid” patient-specific implants”

At the outset, the authors would like to extend their sincere gratitude to the editor and the reviewer for their valuable inputs to the manuscript. The authors have taken all inputs in consideration and have amended the manuscript. Mentioned below you will find the detailed responses to all the comments and suggestions from the reviewers. The authors believe that amendments made in the revised manuscript will be considered satisfactory for publication.

Reviewer 2

This is a small, retrospective study that the authors use to build off of their prior studies.  This study suggests that results are at least comparable to free hand mesh bending.

  • Comment 1:

Who performed the surgeries and who bent the plates? trainees? faculty? one faculty vs multiple faculty?

Author Response:

The authors acknowledge the two valuable comments.

Correction in the revised manuscript:

Line 101-102: Under general anesthesia, a senior maxillofacial surgeon and a resident performed mid eyelid, transconjunctival, and/or transcaruncular approaches to expose the orbital wall defects.

Line 82-84: The orbital floor mesh plate was trimmed, if needed, and hand-molded by a resident specializing in maxillofacial surgery to fit the size of the defect according to the 3D printed orbital model (Fig. 2).

  • Comment 2:

Do you have federal/institutional approval for use of in-house printed models for patient use (i.e. bending of plates) aside from ethics committee approval? Authors do mention briefly in the conclusions medico-legal implications for in-house custom implant printing but do not mention approval for use of these models.

Author Response:

Authors acknowledge the comment and completely agree that it is indeed interesting to know the approvals for use of in-house printed models.

Correction in the revised manuscript:

Line 76-82: The ten subsequent patients received pre-bent plates (MatrixMIDFACE, DePuy Synthes, Solothurn, Switzerland or MODUS, Medartis, Basel, Switzerland) that were produced with equipment provided by the department based on a 3D-printed orbital model by mirroring the non-fractured orbit of the patient using the CE-certified medical software Mimics Innovation Suite v. 20-21 (Materialise, Leuven, Belgium), and two different desktop 3D-printer, a MakerBot Replicator+ (MakerBot Industries, Brooklyn, New York, USA) and an Objet30 Prime (Stratasys, Ltd., Eden Prairie, Minnesota, USA).

Line 82-87: The orbital floor mesh plate was trimmed, if needed, and hand-molded by a resident specializing in maxillofacial surgery to fit the size of the defect according to the 3D printed orbital model (Fig. 2). For intraoperative use, the pre-bent plate was sterilized in an autoclave with a standardized and certified sterilization procedure. During the entire pre-contouring process, the printed 3D model never had contact with the patient. In addition, we used commercially available and medically approved orbital meshes, not directly 3D printed patient-specific implants (PSI).

  • Comment 3:

Authors discuss cost for saved operative time - I would like to see cost estimates for the printing, materials, and time of the person printing (is this 1 hour of physician time?) and compare the differences.

Author Response:

Authors acknowledge the valuable comment and add the cost estimates for the printing material, and time of the person printing. We would like to state that this study primarily aimed to compare the pre- and postoperative volume difference between the absolute volume difference (unfractured orbit – fractured orbit), the fracture area, the fracture collapse, and the effective surgery time between the two groups.

Correction in the revised manuscript:

Line 203-205: The material cost of one 3D printed orbital model in our study is estimated at $2 for PLA (MakerBot Replicator+) and $25 for Med610 (Objet30 Prime). The aforementioned virtual planning of a 3D orbital model took approximately 20 minutes and was usually performed by a surgical resident.

  • Comment 4:

This study discussed trying to improve cost-effectiveness of care by using a hybrid pre-bent implant. However, post-operative CT scans or cone beam CT scans were obtained in patients in the study.  This is not standard in many institutions and increases cost of care and radiation exposure. 

Author Response:

Authors understand and acknowledge the remark from the reviewer. However, in our department post-operative CT scans or cone beam CT scans are standard, but we agree that this increase cost of care.

Correction in the revised manuscript:

-

  • Comment 5:

Authors also mention potential for decreased blood loss and shorter hospitalization time (line 185-6). However, repair of isolated orbital floor fractures are typically outpatient procedures with minimal blood loss.

Author Response:

The authors acknowledge the comment and would like to clarify that in our department orbital floor reconstructions are no outpatient procedures. Our patients are hospitalized after surgery because of the risk of retrobulbar hematoma. We agree with the reviewer that the reconstruction of an isolated orbital floor fracture results in a minimal blood loss.

Correction in the revised manuscript:

Line 194-197: In the study of Zieliński et al a shorter time of operation means lower intraoperative blood loss and expected shorter hospitalization time [26]. In contrast, our study showed no difference between length of stay between the two groups.

Reviewer 3 Report

The paper entitled “Three-dimensional analysis of isolated orbital floor fractures pre- and post-reconstruction with standard titanium meshes and “hybrid” patient-specific implants titanium meshes and “hybrid” patient-specific implants” uses a nice technique to print a model and pre-bend meshes for orbital reconstruction.

Line 70 – please modify the sentence: Twenty-two patients met the inclusion criteria.

Please comment in the text on the in house printing – is this a commercial company or equipment purchased by the department?

Please specify the process of contouring the plate pre-operatively on the model including sterilization of the mesh in the “experimental section”.

Row 188: Using pre-bent implants can considerably reduce operation time by 42.5 min.

Please rephrase: this data pertains to this study… One can argue that contouring a titanium mesh for the orbital floor does not take on average 42.5 minutes… All the rest of the economical calculations are also problematic since they rely on this non-significant data and on subjective conclusions (although equipment for 3D-printing is more expensive, costs can be recuperated through saving surgery time).

Row 191: Please provide an estimate for the cost of printing a model.

Please provide a sentence on the caveats: biased cohort, retrospective analysis etc.

Author Response

Manuscript ID: jcm-787345

“Three-dimensional analysis of isolated orbital floor fractures pre- and post-reconstruction with standard titanium meshes and “hybrid” patient-specific implants”

At the outset, the authors would like to extend their sincere gratitude to the editor and the reviewer for their valuable inputs to the manuscript. The authors have taken all inputs in consideration and have amended the manuscript. Mentioned below you will find the detailed responses to all the comments and suggestions from the reviewers. The authors believe that amendments made in the revised manuscript will be considered satisfactory for publication.

Reviewer 3

The paper entitled “Three-dimensional analysis of isolated orbital floor fractures pre- and post-reconstruction with standard titanium meshes and “hybrid” patient-specific implants titanium meshes and “hybrid” patient-specific implants” uses a nice technique to print a model and pre-bend meshes for orbital reconstruction.

  • Comment 1:

Line 70 – please modify the sentence: Twenty-two patients met the inclusion criteria.

Author Response:

The authors acknowledge the suggestion.

Correction in the revised manuscript:

Line 70: Twenty-two patients (10 women and 12 men) met the inclusion criteria.

  • Comment 2:

Please comment in the text on the in house printing – is this a commercial company or equipment purchased by the department?

Author Response:

Manuscript text amended to reflect the suggestion.

Correction in the revised manuscript:

Line 76-82: The ten subsequent patients received pre-bent plates (MatrixMIDFACE, DePuy Synthes, Solothurn, Switzerland or MODUS, Medartis, Basel, Switzerland) that were produced with equipment provided by the department based on a 3D-printed orbital model by mirroring the non-fractured orbit of the patient using the CE-certified medical software Mimics Innovation Suite v. 20-21 (Materialise, Leuven, Belgium), and two different desktop 3D-printer, a MakerBot Replicator+ (MakerBot Industries, Brooklyn, New York, USA) and an Objet30 Prime (Stratasys, Ltd., Eden Prairie, Minnesota, USA).

  • Comment 3:

Please specify the process of contouring the plate pre-operatively on the model including sterilization of the mesh in the “experimental section”.

Author Response:

Authors acknowledge the comment and completely agree that it is indeed interesting to specify the process.

Correction in the revised manuscript:

Line 82-87: The orbital floor mesh plate was trimmed, if needed, and hand-molded by a resident specializing in maxillofacial surgery to fit the size of the defect according to the 3D printed orbital model (Fig. 2). For intraoperative use, the pre-bent plate was sterilized in an autoclave with a standardized and certified sterilization procedure. During the entire pre-contouring process, the printed 3D model never had contact with the patient. In addition, we used commercially available and medically approved orbital meshes, not directly 3D printed patient-specific implants (PSI).

  • Comment 4:

Row 188: Using pre-bent implants can considerably reduce operation time by 42.5 min.

Please rephrase: this data pertains to this study… One can argue that contouring a titanium mesh for the orbital floor does not take on average 42.5 minutes… All the rest of the economical calculations are also problematic since they rely on this non-significant data and on subjective conclusions (although equipment for 3D-printing is more expensive, costs can be recuperated through saving surgery time).

Author Response:

Authors acknowledge the valuable comment. We replaced the problematic sentence “…equipment for 3D-printing is more expensive, costs can be recuperated through saving surgery time…” through a new sentence in Line 208-210.

Correction in the revised manuscript:

Line 198: In our department using pre-bent implants can considerably reduce operation time by 42.5 min.

Line 201-207: Of course, costs vary greatly depending on staffing levels, hospital infrastructure costs and ultimately the health care system of the different countries. The material cost of one 3D printed orbital model in our study is estimated at $2 for PLA (MakerBot Replicator+) and $25 for Med610 (Objet30 Prime). The aforementioned virtual planning of a 3D orbital model took approximately 20 minutes and was usually performed by a surgical resident. The image segmentation work and fabrication of the 3D model was appreciated by the residents as preparation time for the surgical procedure.

  • Comment 5:

Row 191: Please provide an estimate for the cost of printing a model.

Author Response:

Manuscript text amended to reflect the suggestion.

Correction in the revised manuscript:

Line 203-204: The material cost of one 3D printed orbital model in our study is estimated at $2 for PLA (MakerBot Replicator+) and $25 for Med610 (Objet30 Prime).

  • Comment 6:

Please provide a sentence on the caveats: biased cohort, retrospective analysis etc.

Author Response:

The authors acknowledge the valuable comment and agree that the necessity of reporting the limitations of a study are important. This additional info has been added to the manuscript in the discussion section.

Correction in the revised manuscript:

Line 208-211: This study has certain limitations. First, the sample size is small, but this has no impact to compare the two methods. Second, the study based on a retrospective design with a possible selection bias. The risk for a selection bias is minimal, as we have included all consecutive patients who were treated for orbital floor fracture in our department.

Round 2

Reviewer 3 Report

None